

# Temporal and spatial patterns of small vertebrate roadkill in a supercity of eastern China

Qiong Wu[1], Taozhu Sun[1], Yumeng Zhao[1], Cong Yu[1], Junhua Hu[2] and Zhongqiu Li[1]

[1] Lab of Animal Behavior & Conservation, School of Life Sciences, Nanjing University, Nanjing, China
[2] Chengdu Institute of BIology, Chinese Academy of Sciences, Chengdu, China

## ABSTRACT

An assessment of animal roadkill can help develop road mitigation measures. This article is the first to report data on animal-vehicle collisions (AVCs) in Nanjing, a supercity in eastern China. The research was conducted on a 224.27 km stretch of nine roads in Nanjing. In the period, between November 2020 and October 2021, 26 fortnightly monitoring missions were conducted to gather roadkill carcasses so that we could analyze their temporal and spatial distribution patterns. A total of 259 carcasses were collected, comprising 22 different species, of which 46.42% were mammals and 48.81% were birds. Cats and dogs are the most roadkill mammals, and blackbirds and sparrows are the most roadkill birds. The temporal analysis demonstrated that the peak of vertebrate roadkill occurred from May to July. Spatial analysis showed that the distribution patterns of vertebrate roadkill on different roads varied with a generally non-random distribution and aggregation. By mapping accidents using kernel density analysis, we were able to pinpoint locations that were at high risk for roadkill. Due to the fortnightly survey, our results would underestimate the casualties, even if, our study suggests that the problem of car accidents due to animals should be a cause for concern, and the results of the analysis of temporal and spatial patterns contribute to the establishment of mitigation measures.

## INTRODUCTION

Roads act as barriers, fragmenting habitats just like any other human activity that changes ecosystems, which reduces biodiversity (*Ferreguetti et al., 2020*). The most significant ecological impact of roads, according to *Coffin (2007)* and *Kociolek et al. (2011)*, appears to be traffic-related mortality from animal-vehicle collisions (AVCs, hereafter). Around the world, different taxonomic groups of animals are affected by AVCs, including mammals (*Collinson et al., 2019*; *Ferreira et al., 2014*), birds (*Bishop & Brogan, 2013*), reptiles and amphibians (*Row, Blouin-Demers & Weatherhead, 2007*). Numerous studies conducted over the past few years have produced AVCs results for many different countries. An estimated 340 million birds are killed on U.S. roads each year (*Loss, Will & Marra,*

Corresponding author
Zhongqiu Li, lizq@nju.edu.cn

2014); around 335,000 European hedgehogs are killed on the roads in the U.K. per year (*Wembridge et al., 2016*); and around nine million medium-large mammals are killed in traffic accidents in Brazil every year (*Pinto et al., 2022*). Roadkill data also have been reported from Spain (*Colino-Rabanal et al., 2012*; *Diaz-Varela et al., 2011*; *Lagos, Picos & Valero, 2012*; *Rodríguez-Morales, Díaz-Varela & Marey-Pérez, 2013*), Poland (*Tajchman, Gawryluk & Drozd, 2010*) and Sweden (*Neumann et al., 2012*). Study found that one of the sources of mortality leading to population decline may be road mortality (*Grilo et al., 2012*), which is affecting population size and population structure (*Gibbs & Steen, 2005*).

Previous research on small vertebrates (*Clevenger, Chruszczc & Gunson, 2003*; *Morelle, Lehaire & Lejeune, 2013*) discovered that seasonal behaviors such as dispersal and migration are linked to temporal variations in roadkill patterns. Peak periods of animal roadkill occur during periods when animals are searching for mates or increasing their foraging activity (*Lagos, Picos & Valero, 2012*; *Rodríguez-Morales, Díaz-Varela & Marey-Pérez, 2013*; *Smith-Patten & Patten, 2008*). Also, juvenile dispersal during periods of increased numbers of free-ranging and inexperienced juveniles can affect the seasonality of roadkill (*Madden & Perkins, 2017*). Roadkill patterns exhibit temporal variability, which can be understood to improve mitigation methods, particularly those that can be handled in a timely manner (for example, limitation of traffic intensity or speed) (*D'Amico et al., 2015*).

The spatial distribution of casualties is another key subject in roadkill research (*Jaeger & Fahrig, 2004*; *Trombulak & Frissell, 2000*). According to certain research, roadkill of mammals and other vertebrates, such as birds, does not occur randomly (*Coelho, Kindel & Coelho, 2008*; *Garriga et al., 2017*). Roadkill hotspots are stretches of road where animal-vehicle collisions are more common (*Santos et al., 2015*). Studies indicated that birds are more likely to be road-killed near forests (*Medrano-Vizcaino & Espinosa, 2021*); some mammalian species, such as hares, are more vulnerable near waters (*Freitas et al., 2015*); and snake mortality is higher near pastures and farmlands (*Quintero-Ángel et al., 2012*). These areas might be thought of as top priorities for the use of mitigation measures. By constructing animal crossing structures such as tunnels, ledges in culverts, and overpasses in these locations, we can reduce the numbers of animals that are road-killed (*Grift et al., 2013*).

The world's road network has been expanding steadily in recent years (*Laurance et al., 2014*). In China, the total mileage of the road network and road network density are also increasing. By the end of 2020, the total mileage of China's road network reached 5.20 million km, and the density of the road network in China reached 54.15 km per 100 square km. Among them, the mileage of National highways, Provincial highways and County highways was approximately 0.37 million km, 0.38 million km and 4.46 million km, respectively (from *the Statistical Bulletin on the Development of the Transport Industry in 2021*). Roadkill, which are one of several reasons for the loss of vertebrates, have proven in studies to increase with the development of road networks (*Forman & Alexander, 1998*). Nevertheless, most studies related to animal roadkill have been conducted in ecologically sensitive areas in China, such as the Changbai Mountain Nature Reserve (*Wang et al., 2016*), the Huangnihe River Reserve (*Li, 2019*), the Zoige Wetland National Nature

Reserve (*Gu et al., 2011*), and the Wanglang National Nature Reserve (*Zhang et al., 2018*). However, there are very few studies related to animal roadkill in China's modern cities, and research has only been conducted in Taiwan (*Lin et al., 2019*). Therefore, there is a need to investigate animal roadkill status in a wide geographical space.

The study's aim is to present the initial findings regarding the spatial and temporal patterns of AVCs. We monitored nine road stretches fortnightly for one year in Nanjing (a super-large city in eastern China). The objectives of this study were (1) to describe the species composition of roadkill on the road studied; (2) to describe the seasonal pattern of roadkill in mammals and birds; and (3) to describe the spatial patterns of roadkill on the investigated road. For example, where are roadkill incidents gathered? Where are the roadkill hotspots located? How are roadkill distributed across different spatial scales? Using the temporal and spatial distribution patterns in AVCs, we may also identify targeted hotspots of roadkill to apply mitigating actions.

## MATERIALS AND METHODS

### Study area

The study area (Fig. 1) was located in Nanjing in the southwestern Jiangsu Province (31°14′ to 32°37′N, 118°22′ to 119°14′E). The climate is humid north subtropical with four distinct seasons and abundant rainfall, with annual rainfall averaging 1,106.5 mm. The annual average temperature is 15.4 °C, the highest annual extreme temperature is 39.7 °C, and the lowest is −13.1 °C. Topography is flat, and elevations range from 0 to 753 m, and the average altitude is 35 m. Nanjing is a super-large city in eastern China with a resident population of approximately 9–10 million people. It is also an international integrated transport hub city with a total road mileage of 9,796.325 m and an extensive road network with varying road types and traffic intensities. According to the current Technical Standard of Highway Engineering (JTG B01-2014) in China, roads can be divided into national roads, provincial roads and county roads according to their administrative levels. Considering the time and cost and the area of Nanjing, we selected three national highways, three provincial highways and three county highways, totaling nine roads, as the sampling roads. From Fig. 1, we can also find that these roads pass through eight of the 11 districts in Nanjing, covering a wide area. All nine roads are paved, with the annual average daily traffic volumes ranging from 30,000 to 60,000 vehicles per day on national highways, 10,000 to 40,000 vehicles per day on provincial highways, and 5,000 to 15,000 vehicles per day on rural highways. The types of roadside habitats on these roads cover residential areas, farmland, open fields and forests. The basic conditions of the sampled roads are shown in Table 1.

### Roadkill survey data collection

To gather data on road vertebrate carcasses, we ran 26 monitoring campaigns over 224.27 km of road, totaling 5,831 km, between November 2020 and October 2021. Nine selected roads were monitored every fortnight over the course of a year. Sampling was done in good weather conditions (without rain or snow, which may have resulted in sampling not being carried out on the same day of the week), beginning at 9 a.m. and
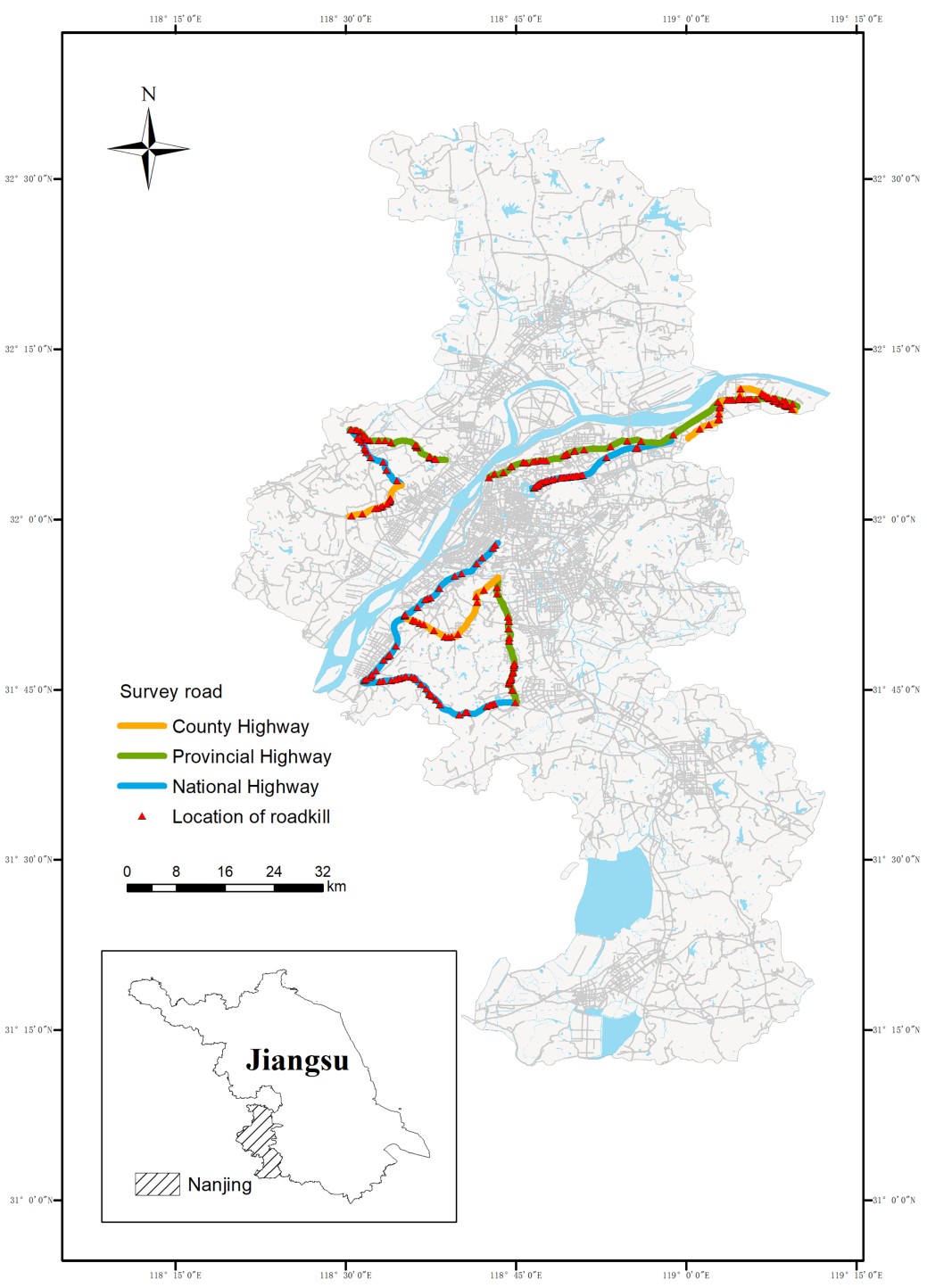

**Figure 1 Sample lines for roadkill survey of small vertebrates in Nanjing.** The 224.27 km road section is in bold, including the National Highway (90.60 km, blue), the Provincial Highway (84.74 km, green) and the Rural Highway (48.93 km, yellow)

lasting as long as it took to complete the entire route while maintaining a speed of no more than 40 km/h (*Ferreguetti et al., 2020*). Considering the safety of driving, our speed should not be too low. The survey team included drivers and observers who kept records of animal

**Table 1 Basic information of the nine sampled roads in Nanjing, China.**

| Road | G1 | S1 | X1 | G2 | S2 | X2 | G3 | S3 | X3 |
|---|---|---|---|---|---|---|---|---|---|
| Highway classification | National highway | Provincial highway | County highway | National highway | Provincial highway | County highway | National highway | Provincial highway | County highway |
| Number of lanes | 8 | 4/6 | 2 | 6 | 4 | 4 | 4/6 | 4 | 4 |
| Presence of road medians | Yes | Yes | No | Yes | No | Yes | Yes | Partly present | Yes |
| Main habitat types on roadsides | Residential areas, open fields | Residential areas, farmland | Residential areas, farmland | Farmland, forests | Residential areas, forests | Open fields, forests | Residential areas, open fields | Residential areas, forests | Open fields, forests |
| Road width | 40~60 m | 20~50 m | 10~20 m | 30 m | 20 m | 15 m | 20~40 m | 15~35 m | 15~30 m |

carcasses found on the survey roads. Additionally, driving recorders were installed in the automobiles to capture any roadkill events throughout the survey. All carcasses found during the survey were identified at class (to species level where possible). The Global Positioning System (GPS) was used for logging the locations of each road accident. Roadkill was georeferenced to an accuracy of 50 m. Once recorded, it was removed from the road. The accuracy of roadkill hotspot detection is decreased due to the low frequency of our sample and the possibility of ignoring small-sized carcasses in car surveys (*Santos, Carvalho & Mira, 2011*; *Santos et al., 2015*). The nine roads sampled passed through a total of eight of the eleven districts in Nanjing, which is a wide survey area. Considering the time, money and labor costs, more frequent surveys were not supported.

## Data analysis

Exploratory analysis was used to calculate the number and species of roadkill animals and their proportion in taxonomic groups and among all roadkill to identify the roadkill composition. We used the Kruskal-Wallis test to examine whether the number of roadkill animals differed significantly between seasons and whether roadkill mortality differed across different roads. The analysis was performed using SPSS 23.

We evaluated the spatial distribution of AVCs in three steps. First of all, we used the nearest neighbor distance (NND) to evaluate the distribution of roadkill on the road network (*Gonser, Jensen & Wolf, 2009*). We compare the observed average distance between each incident and its nearest neighbor with the average distance that would be expected if the accidents were randomly distributed along the road (complete spatial randomness) to determine whether AVCs are aggregated along the road network (*Okabe & Sugihara, 2012*). The Clark-Evans index is calculated to confirm departure from a random distribution. The index is the ratio of the observed and expected mean distances: values >1, =1, <1, indicating that the points are aggregated, randomly distributed and dispersed, respectively (*Clark & Evans, 1954*).

Secondly, to visually describe high-density areas for AVCs, we employed kernel density estimation (KDE). There are two forms of kernel density estimation: planar kernel density estimation (PKDE) (*Loo & Anderson, 2015*) and network kernel density estimation (Net-KDE) (*Okabe & Sugihara, 2012*). PKDE uses Euclidean distance to calculate the density of

points in 2D space, while Net-KDE is used to calculate the density of incidents occurring on the network (*Loo & Anderson, 2015*; *Okabe, Satoh & Sugihara, 2009*). Since the roadkill survey in this study was conducted on roads, we chose Net-KDE to calculate the density of roadkill events occurring on roads. The road network is divided into shorter linear segments (lixel) for aggregation of points on the network. The Net-KDE calculates the density value and assigns it to each lixel (*Xie & Yan, 2013*). Another important issue in computing Net-KDE is the choice of bandwidth, where larger bandwidths tend to ignore local variations (*Khalid et al., 2018*). We experiment with 100 m, 200 m and 400 m bandwidths and 10 m and 20 m lixel sizes. After observing the smoothness of Net-KDE, we found that choosing a bandwidth of 200 m and 20 m pixel size for analysis is appropriate.

Finally, we investigated the spatial structure of AVCs at different spatial scales using the Ripley's K-function, which denotes the pattern of point distribution at various scales (*Bailey & Gatrell, 1995*; *Ripley, 1976*). We applied the network K-function of *Okabe & Sugihara (2012)* as our roadkill points were distributed along the road network. *Okabe & Yamada (2001)* defined the Net-K-function as follows:

$$K(t) = \frac{|L_T|}{n(n-1)} \sum_{i=1}^{n} (\, n(t|pi))$$

where $K(t)$ is the network K statistics, $|L_T|$ is the total length of the road network, $n(t|pi)$ is the number of points $p$ within network distance $t$ of a point $p_i$ averaged over $i$ = 1, 2, ..., n.

Monte Carlo simulation tests the distribution of point patterns on the network and finds the upper and lower critical values of the significance level α (in this study α = 5%). The distribution of points is random if the values of K($t$) lie within the confidence envelope, clustered if the observed values are above the upper envelope and dispersed if they are below the lower confidence envelope (*Okabe & Sugihara, 2012*).

All the spatial analyses were performed with the use of the SANET (Spatial Analysis along Networks) Standalone and ArcGIS 10. SANET Standalone provides tools adapted to perform spatial analysis along linear features.

## RESULTS

### Species composition of roadkill

From November 2020 to October 2021, we examined a total of 5,831 km through 26 surveys. A total of 293 animals (21 identified species) were collected. These included 136 mammals (six species), 143 birds (14 species) and five reptiles (one species) (Table 2). Birds made up 48.81% of the roadkill followed by mammals (46.42%) and reptiles (1.70%).

The most commonly mammalian roadkilled species were the cat (*Felis catus*) (28.7% of mammalian casualties), the dog (*Canis lupus familiaris*) (26.5%) and the weasel (*Mustela sibirica davidiana*) (9.6%). The most common bird roadkilled species were the Chinese Blackbird (*Turdus mandarinus*) (15.4% of bird casualties) and the Sparrow *(Passer montanus)* (7.7%). Together these five species accounted for 41.3% (121) of all roadkill. The only recorded reptile species killed on the road is the Tabby-necked snake (*Rhabdophis tigrinus*).

**Table 2 Frequency of small vertebrates road-killed in the sampling roads of Nanjing, 2020-2021.**

| Common name | Scientific name | N | % of taxa | % of total kills |
|---|---|---|---|---|
| Mammal | | | | |
| Cat | *Felis catus* | 39 | 28.7 | 13.3 |
| Dog | *Canis lupus familiaris* | 36 | 26.5 | 12.3 |
| Weasel | *Mustela sibirica davidiana* | 13 | 9.6 | 3.8 |
| Mouse | *Muroidea* | 11 | 8.1 | 4.4 |
| Hedgehog | *Erinaceus amurensis* | 7 | 5.1 | 2.4 |
| Hare | *Lepus sinensis* | 1 | 0.7 | 0.3 |
| Unidentified mammal | | 29 | 21.3 | 9.9 |
| Total mammals | | 136 | | |
| Bird | | | | |
| Chinese Blackbird | *Turdus mandarinus* | 22 | 15.4 | 7.5 |
| Sparrow | *Passer montanus* | 11 | 7.7 | 3.8 |
| Spotted dove | *Streptopelia chinensis* | 3 | 2.1 | 1.0 |
| Azure-winged magpie | *Cyanopica cyanus* | 3 | 2.1 | 1.0 |
| Light-vented bulbul | *Pycnonotus sinensis* | 3 | 2.1 | 1.0 |
| White-cheeked starling | *Sturnus cineraceus* | 2 | 1.4 | <1 |
| Crested myna | *Acridotheres cristatellus* | 2 | 1.4 | <1 |
| Magpie | *Pica pica* | 1 | <1 | <1 |
| Oriental turtle dove | *Streptopelia orientalis* | 1 | <1 | <1 |
| Chicken | *Gallus gallus domesticus* | 1 | <1 | <1 |
| Bamboo partridge | *Bambusicola thoracica* | 1 | <1 | <1 |
| Red-billed blue magpie | *Urocissa erythroryncha* | 1 | <1 | <1 |
| Red-breasted flycatcher | *Ficedula albicilla* | 1 | <1 | <1 |
| White Wagtail | *Motacilla alba* | 1 | <1 | <1 |
| Unidentified bird | | 90 | 62.9 | 30.7 |
| Total birds | | 143 | | |
| Reptile | | | | |
| Tabby-necked snake | *Rhabdophis tigrinus* | 5 | 100.0 | 1.7 |
| Total reptile | | 5 | | |
| Unidentified body | | 9 | | |
| Total | | 293 | | |

## Seasonal distribution of roadkill

Roadkill occurrences were not evenly distributed throughout the year (Fig. 2), with significant differences among months ($\chi^2 = 20.552$, df = 11, $p = 0.038$) and seasons ($\chi^2 = 12.012$, df = 3, $p = 0.007$). The highest rates occurred in May (32 specimens), June (40 specimens) and July (35 specimens), compared to lower roadkill rates in February (13 specimens), March (15 specimens) and April (15 specimens). The number of roadkill was significantly higher in summer and autumn than in spring and winter. It was also significantly different between seasons for specific taxonomic groups (Mammals:

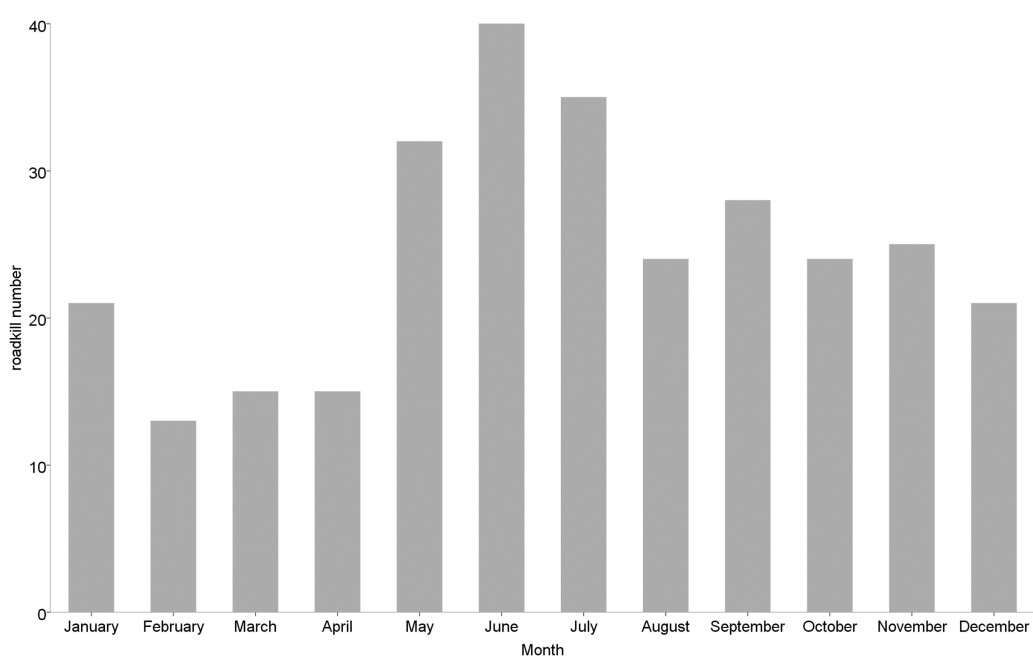

**Figure 2 Monthly number of small vertebrate roadkill in Nanjing, November 2020—October 2021.**

$\chi^2 = 8.726$, df = 3, $p = 0.033$; Birds: $\chi^2 = 11.211$, df = 3, $p = 0.011$). Mammals showed higher numbers of roadkill in autumn. By contrast, birds showed peaks in summer (Fig. 3).

## Description of the spatial pattern of roadkill

Among the different types of roads, provincial roads (5.76 ± 0.65 individuals/100 km) had the highest road mortality, followed by national roads (5.38 ± 0.62 individuals/100 km) and county roads (5.02 ± 0.68 individuals/100 km), but no significant differences in road mortality were found among the different administrative levels of roads ($\chi^2 = 0.471$, df = 2, $p = 0.790$) (Fig. 4).

We observed that carcasses were not randomly distributed along the road network. Among the nine roads investigated, the roadkill of G1, S1, and S2 showed a clustering pattern ($p < 0.05$; Table S1 and Fig. S1). The heterogeneous distribution of AVCs along the road network was revealed by network kernel density analysis. The shade of color in Fig. 5 represented the density of roadkill. The Net-KDE results showed that the darker the road segment, the more road accidents occurred (Fig. 5).

For the multi-distance clustering analysis, the network K function is executed on the road network. The results of the K-function show the clustering tendency of roadkill at multiple distances (Fig. 6). The distribution pattern of vertebrate roadkill on different roads varies, and it is generally non-random distribution and aggregation. On the national highway, the roadkill distribution of vertebrates on G1 and G3 is obviously clustered within 20 km; on G2, it presented a random distribution. On the provincial highway, there was significant clustering of roadkill at small spatial scales. The range of spatial scales over which clustering was significant was 0–5 km in roadkill; on a larger scale, it showed a

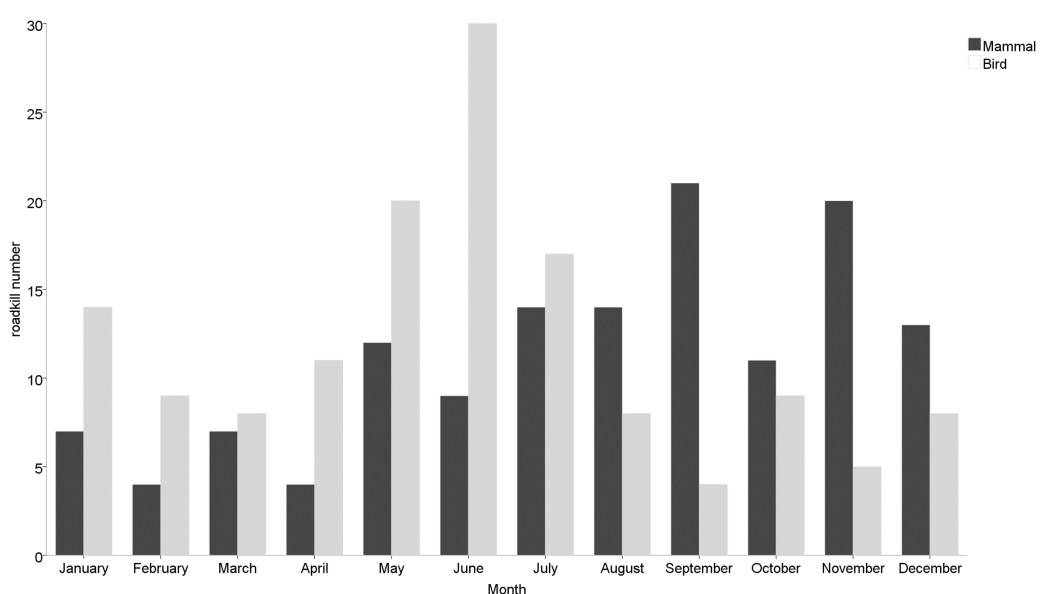

**Figure 3 Monthly roadkill number of mammals and birds in Nanjing, November 2020—October 2021.**

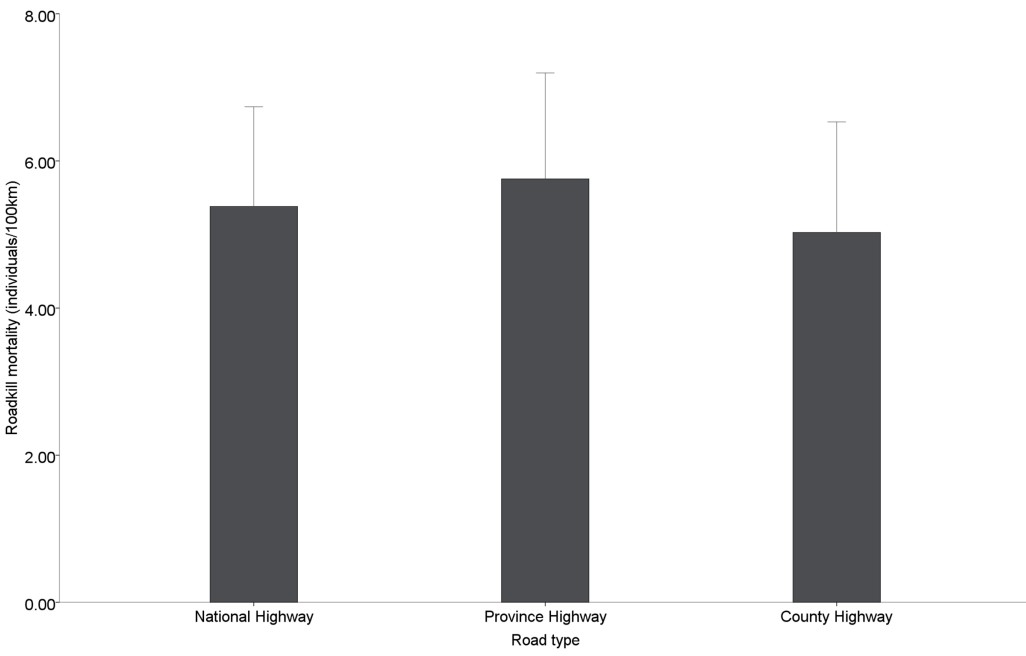

**Figure 4 Roadkill mortality rates on different administrative levels road.**

scattered or random distribution. On the county roads, the distribution of roadkill on X2 and X3 are significantly clustered within 3 km and tend to be randomly distributed with increasing scale. On X1, it is randomly distributed in a small range, tends to be dispersed as the scale increases, and stabilizes at random after 10 km.
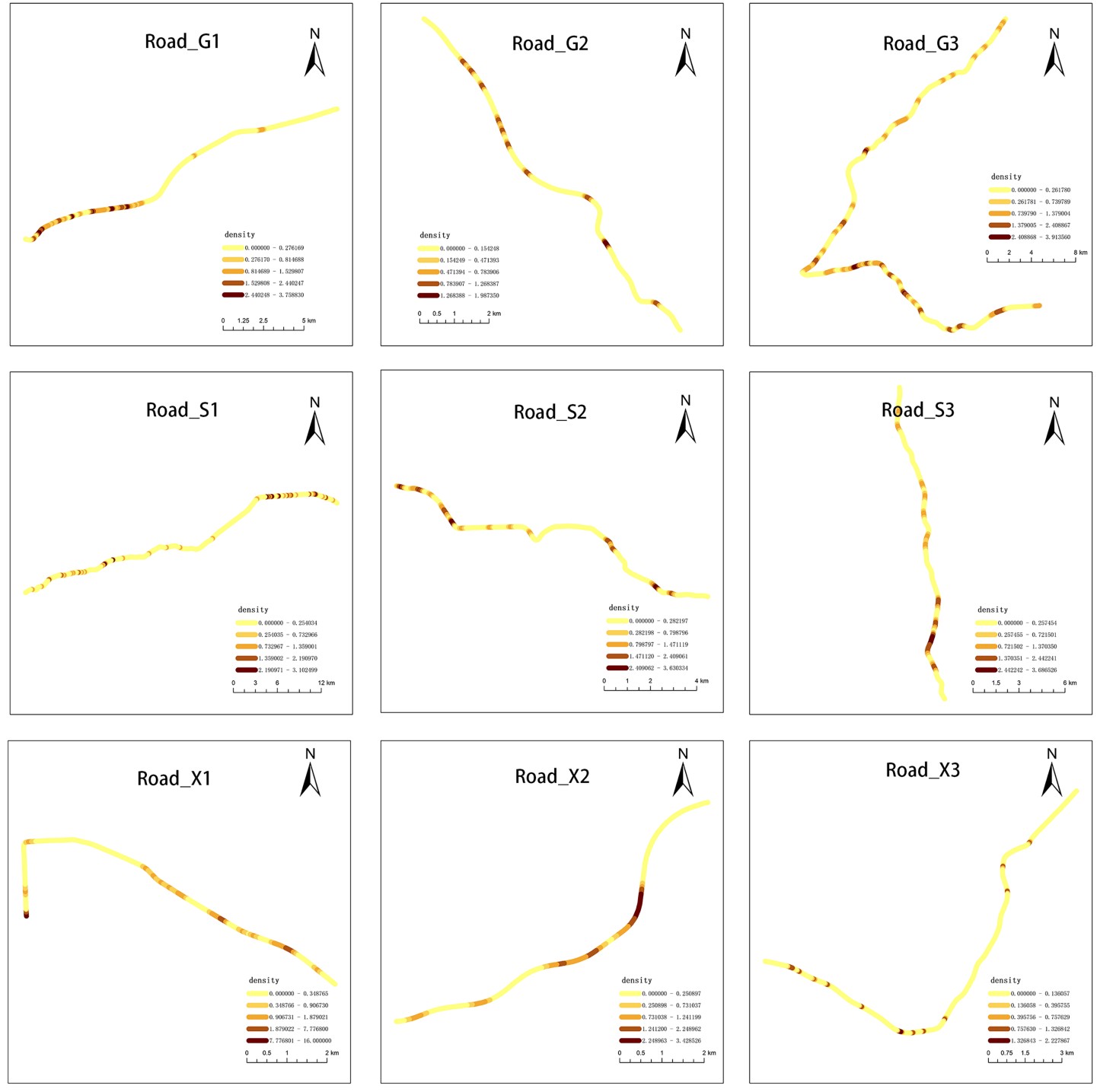

**Figure 5  Results of network kernel density estimation.** Darker colors represent higher roadkills densities.

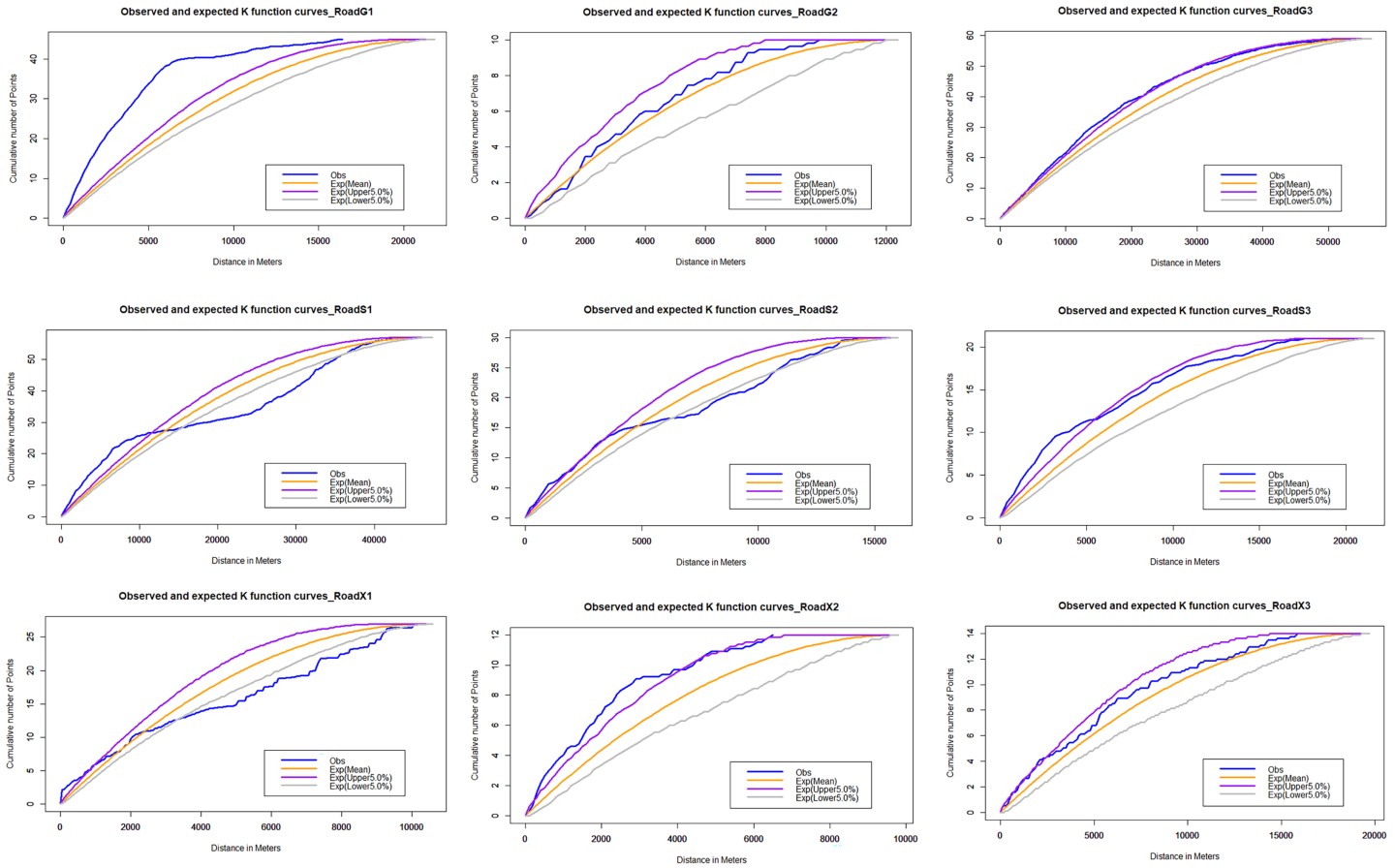

**Figure 6 Network K-function analysis.** The blue line is the observed curve, the orange line is the expected curve, the purple line means the upper limit of the 95% confidence interval and the grey line means the lower limit of the 95% confidence interval. If the observed curve is above the confidence interval, the spatial distribution pattern of AVCs is clustered; below, it is dispersed; within, it is random.

## DISCUSSION

### Roadkill magnitude and species affected

A total of 293 roadkill animals were recorded in our 12-month study, with an average road mortality rate of 5.39 ± 0.37 individuals/100 km. Comparison of vertebrate road mortality across studies is difficult due to differences in survey methods and species conditions at survey sites (*Elzanowski et al., 2009*; *Langen et al., 2007*). However, the number of roadkill in animals has certainly been underestimated in these investigations. Many animals were thrown far from the road after a collision due to the impact, and it was clear that these individuals were often overlooked during surveys (*Santos & Ascensao, 2019*). Alternatively, animal carcasses remained on the road for only a short time due to vehicle movements, rain and the activities of road scavengers (*Santos, Carvalho & Mira, 2011*). Studies have shown this to be an important factor in underestimating road mortality (*Pinto et al., 2022*). In another case of ours, over 90% of vertebrate carcasses remained on the road for no more than 5 days (well below the sampling interval of this study) (Q. Wu & T. Sun, unpublished data, 2022). Research approaches also play an important role in roadkill surveys. The low

frequency of surveys (every fortnightly) may lead to an underestimation of roadkill, as the persistence of bodies on the road is much lower than the survey interval (*Santos, Carvalho & Mira, 2011*). More frequent surveys by foot, both at day and night, can provide more reliable results. In addition, the sampling was conducted in good weather conditions may have resulted in sampling not being conducted on the same day of the week. And traffic levels on different days of the week may have an effect on animal roadkill. For example, *Steiner et al. (2021)* found that higher driving speeds and increased human disturbance (travel, recreation) contributed to the high collision frequency of roe deer on Friday and Saturday.

Roadkill events tend to be concentrated in one or a few species, which are usually locally abundant, mobile and easily attracted by favorable resources or environmental conditions along the road (*Forman, 2000*). Cats and dogs are the most commonly killed mammals on the roads sampled in Nanjing, probably because of their large numbers, wide distribution and high population density (*Yan, 2020*). For example, the total number of owned cats in urban areas of China is estimated to be 149,807,371 (*Li et al., 2021*). The high number of dogs and cats in Nanjing is closely related to the city's high population. In densely populated areas, more people treat cats and dogs as pets and they can accidentally breed, be abandoned by their owners or wander off and become stray animals (*Ozen, Bohning & Gurcan, 2016*).

The blackbird and sparrow are the most commonly road-killed birds, probably because they are both resident birds with large and widely distributed populations and are the dominant species among birds in Nanjing (*Zhang et al., 2018*). Blackbirds are omnivorous birds with a predominantly carnivorous diet. They feed on animals such as flies, maggots and ants around the carcasses of road-killed animals, which is one of the reasons why blackbirds are attracted to the roads. On the other hand, they also feed on grass seeds, plant fruits, *etc.*, thus they are easily attracted into roads by the vegetation along the roadside (*Zhu, 2019*). *Canal et al. (2019)* also found that birds using roadside vegetation strips as 'refuges' may increase their roadkill risk.

## Seasonal distribution of roadkill

Temporal patterns of roadkill may be correlated with the phenology and activity cycles of local animals, as well as local weather circumstances (*Carvalho & Mira, 2011*). Long-term roadkill surveys and interannual roadkill data analyzed in combination with animal life histories make it easier to discover seasonal patterns in animal roadkill. Generally, the peak road kills for mammals appeared in summer and autumn, while the peak roadkill for birds appeared in spring and summer. However, this varies depending on the species' breeding season and dispersal patterns (*Hell et al., 2005*). For animals, their roadkill patterns are associated with three bursts of movement in their life: the mating seasons, the weaning of young to forage with their mothers, and the dispersal of juveniles (*Smith-Patten & Patten, 2008*).

May to September were months in which vertebrates were found to be particularly vulnerable to roadkill. The animals' heightened reproductive activity in summer and autumn may be the cause of the rise in roadkill. In the spring and early summer, many taxa

give birth to their young, and by the end of the summer, the populations have grown significantly (*Raymond et al., 2021*). Not only does this mean more individuals and therefore an increased likelihood of vehicle collisions, but also entails a large number of inexperienced juveniles that must disperse, with an often lacking ability to avoid vehicles (*Legagneux & Ducatez, 2013*). In addition, due to the fact that parents are required to forage more frequently and for longer periods of time to raise their young, some adult animals may also be more susceptible to automobile accidents at this time of year (*Gonser, Jensen & Wolf, 2009*; *Hell et al., 2005*).

Studies have shown that mammalian roadkill peaks occurred in July to September and November, which is probably connected to the duration of the day and the animals seasonal activity (*Tajchman, Gawryluk & Drozd, 2010*). The first peak occurs in association with mammalian breeding activity. From May to September, when most mammals enter the breeding season, increased breeding activity (*Ignatavicius & Valskys, 2018*), mass movements (*Herr, 2008*) and the dispersal of young can all lead to seasonal peaks in roadkill. Conversely, winter is a difficult time for many mammals and corresponds to a period of dormancy or decreased activity for species (*Haigh, 2012*). It is also a time of reduced prey and food supply for many animals and as a result individuals may have to travel further in search of food leading to a greater susceptibility to roadkill that may account for the peak observed in mammals in November (*Garriga et al., 2017*). Moreover, we did not find significant monthly differences in the number of roadkill cats and dogs. It was similar to the investigation results of *Takahashi, Suzuki & Tsuji (2023)* in Ishinomaki City, northern Japan. In urban areas, cats and dogs are kept as pets or cared for by the community as strays. Therefore, cats and dogs use and cross highways more frequently than wild mammals (*Saeki, Johnson & Macdonald, 2007*). Due to human feeding, they can easily obtain enough food every day, and their feeding places and activity ranges are relatively fixed, thus months and seasons may have less influence (*Kays et al., 2020*; *Plantinga, Bosch & Hendriks, 2011*; *Takahashi, Suzuki & Tsuji, 2023*).

The study found that peak roadkill in birds occurred during two periods, from May to July and in January. The period from May to July was a time when birds were engaged in the breeding activity (incubation and fledging) and juveniles were dispersing (*Colino-Rabanal, Lizana & Peris, 2011*; *Erritzoe, Mazgajski & Rejt, 2003*). In particular, the high proportion of immature young individuals leaving the nest may be the cause of the elevated mortality (*Grilo, Bissonette & Santos-Reis, 2009*; *Hell et al., 2005*). On the other hand, during fledging, adults must frequently travel over large distances to forage for food in order to feed their young. As a result, birds may cross roads more frequently or forage among roadside vegetation, which increases the chance of roadkill (*Holm & Laursen, 2011*; *Kuitunen et al., 2003*). In addition, as summer crops mature, seed-eating birds have access to more food sources close to roads, which may potentially be a factor in the increased bird mortality (*Rosa & Bage, 2012*). Meanwhile, we also observed a peak in roadkill in January. It may be correlated with the breeding habits of some bird species and food shortages. Blackbirds and sparrows, for example, can successfully breed and raise two or more broods of young in a year, depending on weather conditions (*Yuan, 2017*; *Zhu, 2019*). Food

shortages could also lead to an increase in animal foraging activity, which could increase the probability of vehicle collisions.

## Spatial distributions

The distribution of the roadkill along the road network is not random, according to the spatial analyses. We indeed observed that most AVCs occurred on national and provincial roads and were highly clustered along these roads in Nanjing, southwestern Jiangsu Province. The study by *Morelle, Lehaire & Lejeune (2013)* in the Wallonia region of southern Belgium also found that although national roads and highways account for small parts of the total length of the road network, more than half of the AVC occur on these roads, and AVCs were highly concentrated in the highway and national roads in the south of Belgium. *Clevenger, Chruszczc & Gunson (2003)* research in Western Canada found that the distribution of roadkill was uneven. Possible explanations for the different patterns are differences in road structure and traffic flow (*Girardet, Conruyt-Rogeon & Foltete, 2015*). In our study area, the average daily traffic volume on national and provincial highways (20,000–50,000 vehicles/day) is two to four times higher than on rural highways (5,000–15,000 vehicles/day) (from Comprehensive Analysis of Nanjing Transportation Economic Operation Report). Although we found higher roadkill mortality on national and provincial roads than on county roads, we did not find significant differences. A possible reason may be that the higher road density of county roads makes animals more likely to cross county roads during movement. *Lao et al. (2011)* found that the road kill points were closer to secondary trunk roads than arterial roads, which increased the road mortality on county roads in some way.

The spatial distribution pattern of roadkill also shows variability between different road types. A map of kernel density revealed places at higher risk of AVCs. This non-random pattern is likely to be explained by road features and land cover (*Gonser, Jensen & Wolf, 2009*). Among our sampled roads, AVCs on G1 are concentrated around Xuanwu Lake and Zijin Mountain National Forest Park. The urban forest park is an important habitat for animals living in cities (*e.g.*, rats, rabbits, weasels, *etc.*), in which there is a wide number and variety of animals (*Fernandez-Juricic, 2004*; *Fontana, Burger & Magnusson, 2011*). The area is also located in the center of Nanjing, with a high population and road density, and large traffic flow. The increased probability of animal crossing the roads in this area makes it prone to animal-vehicle collisions.

The scale of roadkill on different grades of roads is also varies. Network nearest neighbor analysis reveals that the scale of roadkill clusters was higher on national highways than on other roads. On national highways, the scale of roadkill aggregation is in the range of 0–20 km, while on rural highways, the scale of roadkill aggregation is only 0–5 km. Differences in traffic volume, vehicle speeds and road characteristics may be the one of the reasons behind this. *Canal et al. (2019)*, *Williams et al. (2019)* and *Danks & Porter (2010)* showed that traffic intensity was associated with animal road kill. Roads with many curves and low traffic speeds may result in more discontinuous roadkill patterns (*i.e.*, roadkill aggregation on smaller spatial scales) than high-speed linear highways with high speeds and little speed variation (*Clevenger, Chruszczc & Gunson, 2003*).

## CONCLUSIONS

In our study over a 12-month period from November 2020 to October 2021, we collected 293 roadkill individuals along the sampling roads in Nanjing. Temporal analysis of roadkill incidents showed that peak roadkill occurred in the summer and autumn for birds and mammals, respectively. Spatial analysis of roadkill events showed that their spatial distribution pattern on roads was generally non-random and aggregated. The study suggests that animal-vehicle collisions should be taken seriously and the relevant authorities should take necessary mitigating measures. For example, temporary mitigation measures, such as speed limits and temporary traffic control, can be implemented during peak periods of AVCs, taking into account the life history of the animal. For animals such as cats and dogs that frequently use the roads, animal fencing and passages can be set up to prevent animals from entering the roads and direct them to crossing structures as well as to avoid interfering with traffic. Furthermore, long-term monitoring of mitigation after implementation to determine the effectiveness of these measures and timely adjustments based on actual conditions is conducive to minimizing roadkill.

## ACKNOWLEDGEMENTS

We thank Sihan Ning, Yunshu Wang and Lixin Chen for their help during the road surveys. We are grateful to Qin Zhu, Yigui Zhang and Yunchao Luo for their advice on the work. We appreciate Changjian Fu and the editors and reviewers for their valuable comments and suggestions for improving the manuscript.

### Funding

Qiong Wu, Taozhu Sun, Yumeng Zhao were supported by the National Key Research and Development Program of China (2022YFC3202104) and the Natural Science Foundation of Jiangsu Province (BK20211151). There was no other external funding received for this study. The funders had no role in study design, data collection and analysis, decision to publish, or preparation of the manuscript.

### Grant Disclosures

The following grant information was disclosed by the authors:
National Key Research and Development Program of China: 2022YFC3202104.
Natural Science Foundation of Jiangsu Province: BK20211151.

### Competing Interests

The authors declare that they have no competing interests.

### Author Contributions

- Qiong Wu conceived and designed the experiments, performed the experiments, analyzed the data, prepared figures and/or tables, authored or reviewed drafts of the article, and approved the final draft.

- Taozhu Sun performed the experiments, analyzed the data, prepared figures and/or tables, and approved the final draft.
- Yumeng Zhao performed the experiments, prepared figures and/or tables, and approved the final draft.
- Cong Yu performed the experiments, prepared figures and/or tables, and approved the final draft.
- Junhua Hu analyzed the data, authored or reviewed drafts of the article, and approved the final draft.
- Zhongqiu Li conceived and designed the experiments, analyzed the data, authored or reviewed drafts of the article, and approved the final draft.

## Data Availability

The raw measurements are available in the Supplemental File.

## Supplemental Information

Supplemental information for this article can be found online at http://dx.doi.org/10.7717/peerj.16251#supplemental-information.

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
