# Peer review of "Temporal and spatial patterns of small vertebrate roadkill in a supercity of eastern China"

_PeerJ, doi:10.7717/peerj.16251_

## Round 0.1 · original submission · Major Revisions

Please revise your paper to address the concerns of the reviewers.

Reviewer 1 ·

Basic reporting

.

Experimental design

.

Validity of the findings

.

Additional comments

Overall opinion

The authors made 26 trips and counted dead mammals and birds near a particular city in China. They counted 293 animals, and found that they were mostly the common species of the area, that they found more cases when the animals were more active and near a park where humans and fauna coincide. In other words, everything they found is to be expected with that kind of sampling. The problem with this type of study is that it severely undersamples the casualties (as mentioned, for Example, by Pinto et al. 2022, who the authors cite).
The paper is of local interest to Chinese administrators who might want to reduce the problem. It might reach more interested readers in a local journal, but it is still correct and only need some improvements that can be easily done.

Specific recommendations

Improve the discussion by more critically addressing the methodological problem of the study (only frequent sampling by foot, at night and day, can provide reliable results. Counting animals only at daytime, and from a car moving at 40 km/h, misses most cases, particularly for birds and most mammals, which are much smaller than the cats and dogs that you report here).
How does the method problem affect your results? Your conclusions?
I also recommend improving the abstract, by mentioning the method used, the methodological problems, and concrete recommendations. Remember that only properly enforced speed reduction and road closures during seasonal migrations have proven to be effective. All other measures that you suggest are not successful.

Reviewer 2 ·

Basic reporting

There are a few minor issues with the accuracy of the English used in the article which I have flagged in my attached PDF.
The introduction and background to the study is sufficient, though there is at least one example where a literature reference has been misused/misinterpreted in the Discussion (see further details in attached PDF comments).
The structure of the article is appropriate, though there are a couple of figures (Figure 6 and Figure S1) that need better explanations in their legends, and also could be made clearer by annotating or circling the most important areas of the graphs.

Experimental design

The authors have identified the knowledge gap being investigated in this study, and have supported this with their results. In this case, the primary strength of the paper is the collection of empirical roadkill records from a previously under-studied region of China. The main issues with this paper is the lack of detail in the methods and the lack of depth in the analysis. The methods are not reproducible in the current condition. This is particularly relevant for the ‘data analysis’ section. Additionally, although the authors clearly put in notable effort collecting the data every fortnight for a year, trying to investigate seasonal variation in roadkill risk is limited by only one year of data – we do not know if this is a ‘typical’ year or whether these seasonal patterns would be consistent with other years.

Validity of the findings

Raw data has been provided in a supplementary table. The conclusions are mostly appropriate; however, some inferences are not supported by the results e.g. concluding that ‘Differences in vehicle speeds and road characteristics are one of the reasons why it occurred’ (line 343), when the authors did not statistically test for the effects of vehicle speeds and road characteristics on roadkill occurrence. Additionally, the authors aimed to identify where road mitigation could be implemented but they have not actually identified in their results where these locations should be – they have highlighted areas of high roadkill occurrence but haven’t stated which areas should be prioritized for mitigation. This may simply need some rewording so that the aims are less explicit about identifying where to put mitigation.

Additional comments

Overall, this paper provides an insight into the spatial and temporal variation in roadkill across an infrastructure-dense area in China. They provide a simplistic look at roadkill clusters or hotspots, identifying areas where the most roadkill occur, but do not include/account for any environmental variables or landscape factors that might influence the location of the hotspots. The overall concept and methodology are suitable for investigating roadkill; however, the methodology is in general quite vague and the data analysis is currently not reproducible. Additionally, the inclusion of domestic pets in the results is questionable, considering the paper sets out to focus on wildlife conservation, and the inclusion of cats and dogs no doubt heavily influences the patterns that they then go on to describe. There are also a couple of instances where conclusions have been inferred without having been directly tested in this study. A number of changes need to be made to improve the clarity of this manuscript before it is acceptable for publication.

Annotated reviews are not available for download in order to protect the identity of reviewers who chose to remain anonymous.

·

Basic reporting

The English is overall reasonable and clear. However, the paper includes writing in the first person, such as “we” and “our”. I am of the opinion that that is not suitable for this overall clear and concise paper.

I notice that the authors are referring to wildlife-vehicle collisions (WVCs) (such as in Line 50), which could perceptively exclude domestic animals. Therefore, it is advisable that the authors use the term animal-vehicle collisions (AVCs) (which is more inclusive).

The authors often write “roadkill”, “road kill” and “road-kill” throughout the paper. Only one of these should be written and used for the whole paper.

The previous research is mostly well-cited but there were some key challenges I found, that can be easily corrected. The authors could be advised to expand on the relevance of the previous studies that have been conducted elsewhere (Lines 52 to 53). I found that the paper could especially benefit from explaining the knowledge gap a little more clearly. I understand that the study is the first to be conducted in the supercity of Nanjing. However, the paper should explain why this is relevant to scientific communities, road management authorities, etc. They should focus explaining these gaps in the paragraphs between lines 69 to 96.

The structure of the paper conforms to the standard article structure. Regarding Figures and Tables, I only found Figure 1 to require a minor improvement; the authors need to insert a North Arrow on their map.

Thank you for providing the raw data. All raw data that were made available are in accordance with the data sharing policy and are sufficiently comprehensive.

The submission is self-contained and represents an appropriate unit of publication. All relevant results have been included.

Experimental design

This research falls within the aims and scopes of the Journal it has been submitted to.

The research question has been well-defined and is meaningful for the scope of the study. However, the authors should consider explaining more clearly how this study fills the identified knowledge gap (see earlier comment in 1. Basic Reporting). Indeed, the authors should try to identify this gap more clearly and highlight its relevance. I can understand idea and intention of the paper, it just needs to be communicated more effectively.

The experimental design has been well-structured, and the research questions were outlined clearly by the researchers. The authors have conducted a rigorous investigation and upheld high ethical standards.

The methods have overall been described with sufficient information. I only have a suggestion (Line 162), where the authors explain Kernel Density Analysis. I feel that this can be explained better with one or two sentences.

Validity of the findings

The findings are valid. Decisions have not been made based on any subjective determination of impact, degree of advance, or being of interest to only a niche audience.

All underlying data have been accordingly provided in an acceptable, accessible format. They are robust, statistically sound and controlled. The authors have stated the average traffic volume in Lines 322 to 324, but do not reference this. If they have no raw data to confirm these statistics, they must at least provide a reference.

The conclusions have been well-stated, clear and concise. They are explicitly linked to the research question and are limited to the supporting results. All claims and findings have been appropriately supported with relevant research and well-controlled experimental intervention.

Additional comments

This is generally a good paper. Although I have made some extensive comments for revisions that is not at all taking away the significant efforts the authors have put into writing a relevant and concise work. There is great potential for this paper to reach a wide audience, however, the authors must improve on the writing and language, as well as communicating the relevance of this work.

The knowledge gap requires further emphasis and explanation and the authors have been advised to make the effort of addressing this.

There should be enough consistency between writing specific terms, such as “roadkill”.

I have included some detailed corrections from the reviewed word document for this paper and highly encourage the authors to read through those comments.

---

## Round 0.2 · Minor Revisions

Please make the minor revisions suggested by Reviewer 2.

Reviewer 1 ·

Basic reporting

.

Experimental design

.

Validity of the findings

.

Additional comments

In my opinion, the authors made good use of the reviewers' recommendations and the manuscript is now fit for publication, after the standard in-house format review that I am sure you do.

Reviewer 2 ·

Basic reporting

Methods still lack some clarity in places. Terminology sometimes still needs correcting e.g. 'roadkilled' and 'roadkills' are not words and should be replaced. Some areas of discussion are not relevant or need rewording. See 'Additional Comments' for more details.

Experimental design

No comment.

Validity of the findings

No comment.

Additional comments

Methods
• Authors have responded to my query about why these roads were chosen in particular but then have not added that information to the manuscript. Please add justification of road choice to the methods in the manuscript.
• ‘Sampling was done in good weather conditions’ – please specify in the methods in the manuscript that sampling was not carried out on the same day of the week, and then add a sentence or two discussing the potential implications of this in your discussion
• Methodology for KDE still hasn’t been expanded. My original comment stating that the description was vague was an indication that there needs to be more information in the methods about HOW the authors carried out KDE (not just what KDE is)
• Information about ‘the K-Function method was applied in this study using Monte Carlo simulation to find the upper and lower critical values…’ in the authors’ response to reviewers (point 15 in the previous review) should be specified in the methods
Results
• Still not convinced that the authors can claim seasonal patterns when there is only one year of data; however, if they do call it ‘seasonality’ or ‘seasonal distribution’ in the results, then they need to have a sentence or two in the discussion about the limitation of estimating seasonal variation with just one year of data

Additional comments
• ‘Roadkilled’ and ‘roadkills’ should be replaced
• Line 144 – ‘the study’s aim is to present Nanjing with the initial findings…’ – this rather limits the scope of the study to a very localised area and almost suggests it should be published in a more localised journal
• Line 209 – between seasons or months or both?
• Line 337-351 – paragraph about blackbirds – poorly worded.
• Line 361 – ‘for animals, their roadkill patterns are associated with three bursts of movement in their life: the mating seasons, the weaning of young to forage with their mothers, and the dispersal of juveniles’ – do these still apply to your two most dominant species: domestic cats and dogs?
• Lines 365-377 – ‘In this study, overall roadkill distribution was seasonal…’ – Need some mention here, however, that this study may not be representative of longer-term seasonal patterns because the data was only collected during one year so is only representative of that year
• Lines 442-453 – paragraph starting ‘The spatial distribution pattern of roadkill also shows variability…’ – again, this paragraph is focusing on wildlife species and factors affecting wildlife mortality, yet your two most common species found as roadkill are domestic dog and cat
• Mitigation measures – this whole section could be removed because you don’t actually include any analysis of mitigation measures in your study, nor attempt to predict where mitigation would be most effective. It’s fine to have a comment in the conclusion (as you have done) about how your study can contribute to understanding where mitigation should go, but I don’t think you need this whole paragraph on the topic.

---

## Round 0.3 · accepted · Accept

Thank you for making your revisions.